# Parental mediation and the use of social networks: A systematic review

**David Sevilla-Fernández**[1], **Adoración Díaz-López**[2], **Vanessa Caba-Machado**[2], **Juan Manuel Machimbarrena**[3], **Jessica Ortega-Barón** [4], **Joaquín González-Cabrera** [2]*

1 Faculty of Education, Universidad Internacional de La Rioja (UNIR), Logroño, Spain, 2 Instituto de Transferencia e Investigación (ITEI), Universidad Internacional de La Rioja (UNIR), Logroño, Spain, 3 Faculty of Psychology, University of the Basque Country (UPV/EHU), Donostia, Spain, 4 Faculty of Psychology, University of Valencia (UV), Valencia, Spain

* joaquin.gonzalez@unir.net

## Abstract

Social networks are used daily by almost all adolescents and young people. They are used extensively, and their positive aspects are well-known, but they present multiple risks. Parents' mediation (PM) in their children's use of social networks (SNs) to prevent numerous problems has been recently researched. This systematic review analyzed works published since 2012 about online PM strategies concerning the use of SNs by children/adolescents aged between 9 and 18 years old. Following the PRISMA protocol, this review was submitted to Prospero (ID: CRD42022345033), five electronic databases were examined (WOS, SCOPUS, ERIC, ProQuest Psychology, and PubMed), and to assess the risk of bias the ROBIS tool was used. 32 papers were selected. The results indicate that PM strategies (especially the restrictive strategies over the enabling ones) effectively reduce the time spent on SNs and their associated risks. However, it is unclear whether one type of strategy is more effective or whether a combination of the different strategies, depending on the problem to be addressed, is most effective. Finally, no studies have been found that have analyzed the relationship between PM strategies and the benefits associated with the use of SNs. Possible lines of action for future programs and research are proposed.

## 1. Introduction

Relationship, information, and communication technologies have become a vital resource in the daily life of young people. They are necessary to obtain information and entertainment, to study, communicate, and, particularly, to relate to their peers [1]. Their use begins at increasingly younger ages [2]. In this regard, a report by the Pew Research Center [3] indicates that 60% of children under 12 began interacting with a smartphone before age 5. In addition, almost one in five children under 12 own a smartphone [3]. Along similar lines, during adolescence, there is a massive consumption of mobile devices [4], whose use has increased exponentially after the COVID-19 pandemic [5, 6], becoming the preferred device for adolescents to access online leisure in general and social networks (SNs) in particular [4].

**Data Availability Statement:** It has been decided to create a repository in zenodo.org [https://bit.ly/repositorio_rs] where the following documents will appear: • Algorithm, search data. • Protocol

registered in PROSPERO. • Table of application of the inclusion/exclusion criteria of the 303 articles that were screened.

**Funding:** This study was funded by Universidad Internacional de La Rioja [(UNIR Research Plan (2020-2022 and 2022-2024)]. Ministerio de Ciencia, Innovación y Universidades (PID2023-147754NB-I00). The funders had no role in study design, data collection and analysis, decision to publish, or preparation of the manuscript.

**Competing interests:** The authors have declared that no competing interests exist.

SNs are defined as a service that allows users to build a public or semi-public profile within a delimited system, articulate a list of other users or contacts with whom they relate and share common interests, and be aware of their connections with other users and of other platform users with each other [7]. They are used by 4.8 billion people worldwide [8]. Adolescents' use of SNs is a normative experience because 98.5% of young people have some profile in SNs, and 83.5% are registered in more than three SNs [4]. Although most platforms have age restrictions, they are also used by those under 12 [3]. Thus, investing time in SNs, along with chatting and listening to music, is the most frequently performed activity by young people on the Internet [4]. Taking all this into account, this new online scenario can potentially influence minors' psychological and social development [9].

On the one hand, SNs play an essential role in young people's lives, providing them with important benefits such as perceived social support [10], the ease of creating online communities, searching for and transmitting information, feeling more accepted, developing creativity, increasing communicative frequency, or breaking down social boundaries [8]. They can also be beneficial in reducing isolation, improving social skills, and providing a platform for continuous communication and maintaining friendships when separated by physical distance [11]. In addition, mental health professionals acknowledge that the communication and connectivity of SNs could be especially helpful for young people with mental health problems, who are more vulnerable to isolation [12].

However, SNs can also be the main window to several risks of the Internet, as they are currently one of the primary spaces for minors' socialization [1]. Moreover, most of them can be used by minors without needing to attach an identification document, which allows anonymity or identity theft [13]. Their use has been linked to mental health issues such as anxiety, depression, and low self-esteem [14]. In this sense, various studies relate SNs to risks of a dual nature: dysfunctional and relational [15]. Dysfunctional risks refer to the inappropriate use of technology, leading to problematic (even addictive) use, whereas relational risks refer to problems of victimization and violence mediated by third parties. Hence, some of the dysfunctional risks associated with SNs are Fear of Missing Out (FoMO), defined as the fear of being left out of activities that are pleasurable to others [16]; social media disorder, understood as the homonym of Internet Gaming Disorder but applied to SNs [17]; social media fatigue, considered mental load or fatigue associated with the excessive use of SNs [18]; disclosure of private information; mental health issues [19]; and body image distortion [20], among others. Concerning relational risks, cyberbullying [21], minors' solicitation and sexualized interaction with adults [22], or sexting, especially non-consensual and coercive sexting [23], are noteworthy.

Thus, the common exposure factor between the benefits and risks of SNs is the usage time of these platforms. In this sense, the time spent on SNs is not a risk in itself. Still, it is important to analyze usage time because, on the one hand, it can explain part of the variance of risks and benefits, and on the other hand, it is one of the main variables measured regarding digital media [24, 25].

However, most adolescents use digital media mainly within the family environment, where parents moderate the use of SNs [26]. How parents exercise that role through different behavioral strategies has been called Parental Mediation (PM). Livingstone et al. [27] define PM as families' efforts to maximize opportunities and minimize the risks of Internet use in their children. The evolution of the study of PM from television [28, 29] to the current development of the Internet of Things has given rise to different typologies. Thus, various authors have proposed different taxonomies to classify the mediation strategies used by parents in the last two decades. In this sense, the classical taxonomies of Nathanson [28] or Valkenburg et al. [29] suggest three dimensions: active mediation, restrictive mediation, and co-viewing/co-use. Over time, new typologies have appeared based on new technological developments

and advances in research. For instance, Livingstone and Helsper [30] propose four categories: Active co-viewing, restrictive interactions, technical restriction, and monitoring. Other authors like Lwin et al. [31] combine the restrictive and active types in four categories (restrictive, promotive, selective, and laissez-faire), whereas Kirwil [32] distinguishes five strategies: technical restriction, social co-viewing, time restriction, website restriction, and non-restrictive rule-making. Along the same lines, a few years later, Nikken and Jansz [33] suggest an approach encompassing active mediation, co-use, general restrictive mediation, content-specific restrictive mediation, and supervision. In this mosaic of taxonomies, other authors combine the concepts of PM and parenting styles to propose their tools for evaluating PM, like Valcke [34] or Valkenburg [35]. The variability of classifications has influenced the diversity of the measurement tools used and the plurality of the results obtained [36, 37]. However, one proposal to reduce the aforementioned mediation categories has received empirical support, grouping them into a dichotomous categorization of two macro-categories [27]. On the one hand, enabling mediation (EM) includes active mediation strategies for Internet safety, active mediation of the use of the Internet, and technical mediation and monitoring. On the other hand, restrictive mediation (RM) refers to strategies to limit minors' access to the Internet or their online activity.

It seems clear that PM as a whole is considered one of the most effective methods to address the risks faced by young people on the Internet [26, 30, 38–40]. In this sense, throughout the literature, these PM strategies have been related to different behaviors on the Internet. Thus, meta-analysis studies report that RM was more effective than active mediation in decreasing the time children spent on SNs [41]. In addition, RM effectively reduces the disclosure of personal information and online communication with strangers [42, 43] and decreases risky online behavior [44]. RM is also related to a lower risk of cybervictimization [37]. On the other hand, reviews of EM report that it was more effective than RM in decreasing Internet usage time and the likelihood of engaging in risks such as cyberbullying [37, 41] or contacting strangers and disclosing private information [45]. Moreover, it was observed that EM decreased the likelihood of being a perpetrator of online cyberbullying [46] and the risk of mental health issues [19, 47].

Despite the relevance of the role of PM in the use of the Internet and the numerous reviews carried out in this regard [39, 40, 45], to date, except for the approximation to the state of the art by Beyens et al. [48], few efforts have systematically addressed the specific relationship between PM and SNs. SNs are a scenario in which young people can find numerous benefits but also various risks [49], and PM can influence their use in both directions [27]. Therefore, in this context, PM is presented as a protective factor against the risks and an enhancer of the benefits of SNs. In addition, another aspect to consider is the heterogeneity of PM categorizations already discussed, the plurality of uses, and the number of problems associated with SNs. It is necessary to order the results obtained in this field to shed light on the role of PM in young people's use of SNs and to determine whether, as pointed out by Livingstone et al. [27], PM maximizes the opportunities and minimizes the risks of SNs.

Considering the above, this paper aims to answer the following research questions:

## 2. Method

### 2.1. Protocol and eligibility criteria

A systematic review of research published between January 1, 2012, and December 18, 2023 (i.e, the last 10 years) was conducted. Determining the exact lower limit of a time range is complicated and arbitrary. We therefore decided to set this period because the widespread and massive use of social networks between teenagers and young adults started in 2012 with the

**Table 1. Eligibility criteria according to PICOS framework.**

| PICOS Framework | Eligibility criteria |
|---|---|
| Population | Children and adolescence from 9 to 18 years old, both included. |
| | Information collected about parents' mediation strategies and the use of SN sites must have been obtained from the children, their parents/guardians, or both (parents and children). |
| Intervention | N/A |
| Comparison | N/A |
| Outcome | Studies that provide quantitative information, both on PM strategies in any of their taxonomies (active, restrictive, controlling, monitoring, enabling, etc.) and on the use of SNs by children and adolescents (from 9 to 18 years old, both inclusive). |
| Study design | Peer-reviewed empirical studies with cross-sectional or longitudinal designs that relate PM strategies to the use of SNs by children and adolescents. |
| | Samples from the chosen studies could have been selected using any sampling technique or sample size |

Note: N/A = Not applicable

creation of Instagram and later, in 2016, Tik Tok [50]. Furthermore, it was not until beginning of 2012, when online parental mediation become a line of interest for scientist and started to be investigated in relation with other variables. 32 manuscripts that met the eligibility criteria. The PRISMA guidelines were followed [51, 52]. This review was submitted to Prospero (ID: CRD42022345033) available in: https://www.crd.york.ac.uk/prospero/display_record.php?ID=CRD42022345033. Five electronic databases were examined, among the most recognized in the field of science in general (WOS, SCOPUS) and in psychology and education in particular (ERIC, ProQuest Psychology, and PubMed).

All selected articles met the following inclusion and exclusion criteria according to the PICOS framework (Table 1). Firstly, the search was limited to original, peer-reviewed empirical studies with cross-sectional or longitudinal designs that relate PM strategies to the use of SNs by children and adolescents, excluding studies that only provide information on PM strategies, on the use of SNs, or do not relate the two variables. Secondly, studies were included that provide quantitative information, both on PM strategies in any of their taxonomies (active, restrictive, controlling, monitoring, enabling, etc.) and on the use of SNs by children and adolescents (from 9 to 18 years old, both inclusive). Studies using only qualitative methodology, reviews or theoretical studies, or studies referring to offline SNs were excluded. The use of SN sites could be general (without alluding to one or more SNs in particular) or specific (measuring a particular SN site or sites). Thirdly, information collected about parents' mediation strategies and the use of SN sites must have been obtained from the children, their parents/guardians, or both (parents and children). Fourthly, samples from the chosen studies could have been selected using any sampling technique or sample size. Finally, papers published in languages other than English or Spanish were excluded.

## 2.2. Search strategies and study selection process

To define the search terms, firstly, the main topics related to PM and the use of SNs were established. In this sense, the different ways of naming these two major constructs were contemplated to access all the articles and start from a general theme plan. Next, a preliminary literature search was carried out in two databases (WOS and SCOPUS) to locate the keywords included in the scientific articles and the terms used in their titles. The descriptors with the highest frequencies and terms of the thesaurus related to the proposed topic were selected.

Table 2. Search field, filter and result of the different databases.

| DATABASES | SEARCH FIELD | FILTER | RESULT |
|---|---|---|---|
| Web of Science | Topic (Title, Abstract and Indexing). | Type of document: Scientific article.<br>Date of search.<br>Language: English/Spanish. | 255 |
| Scopus | Title, Abstract, Keyword. | Type of document: Scientific article.<br>Date of search.<br>Language: English/Spanish. | 218 |
| PudMed | Title, Abstract | Date of search.<br>Language: English/Spanish. | 87 |
| ProQuest Psychology | Title, Abstract | Type of document: Scientific article.<br>Date of search.<br>Language: English/Spanish. | 19 |
| ERIC | Title, Abstract | Type of document: Scientific article.<br>Date of search.<br>Language: English/Spanish. | 2 |

Finally, the following search algorithm was created: ("parental mediation" OR "parental media mediation" OR "parental digital mediation" OR "parental monitoring" OR "parental control" OR "parental supervision" OR "active mediation" OR "restrictive mediation" OR "enabling mediation" OR "co-using") AND ("social media" OR "social networking" OR "social network" OR "social networks" OR "SNS" OR "online community" OR "social platform" OR "social media platform" OR "social platforms"). We try to maintain the same filters and search fields in all databases, although the available options are not always the same (see Table 2).

Two independent researchers conducted an unblinded, standardized, and independent eligibility assessment. The database search protocol identified a total of 581 publications as potential documents. Correctly managing duplicate studies is essential for a systematic review. Therefore, we followed the recommendations of Kwon et al. [53] and eliminated the duplicate studies found in the different databases after careful control by two of the researchers, using the ZOTERO bibliographic reference manager. After the initial screening stage, 278 references were discarded, and 303 publications reached the eligibility stage. Due to the numerous variables related to the target variables and the studies' methodological and conceptual variability, each researcher completed a review of the 303 documents to screen them according to the inclusion and exclusion criteria. Disagreements about the inclusion/exclusion of the selected articles were resolved after consulting a third investigator and discussing their eligibility until reaching 100% agreement. Articles that did not meet the inclusion criteria were eliminated from the study. Finally, 32 articles were selected for the present systematic review.

The summary of the pre-selected and selected articles at each stage of the selection process was represented in a flowchart (see Fig 1).

### 2.3. Risk of bias

There is no doubt that bias in systematic reviews (SR) is almost inevitable. Although we have tried to minimize it, there can always be some subjectivity in the (SR) process. The main risks of bias can be grouped into: eligibility criteria of the studies, identification and selection of the studies, data collection and assessment of the studies, and synthesis and conclusions. To reduce these risks, the ROBIS tool was adapted to our research work [54]. First, we ensured that the eligibility criteria were relevant, well-defined and sufficiently restrictive based on the objectives of our research and the sources of information. Second, the two main authors, who performed the analyses, oversaw this task independently and agreed 96.7%, and the Cohen´s

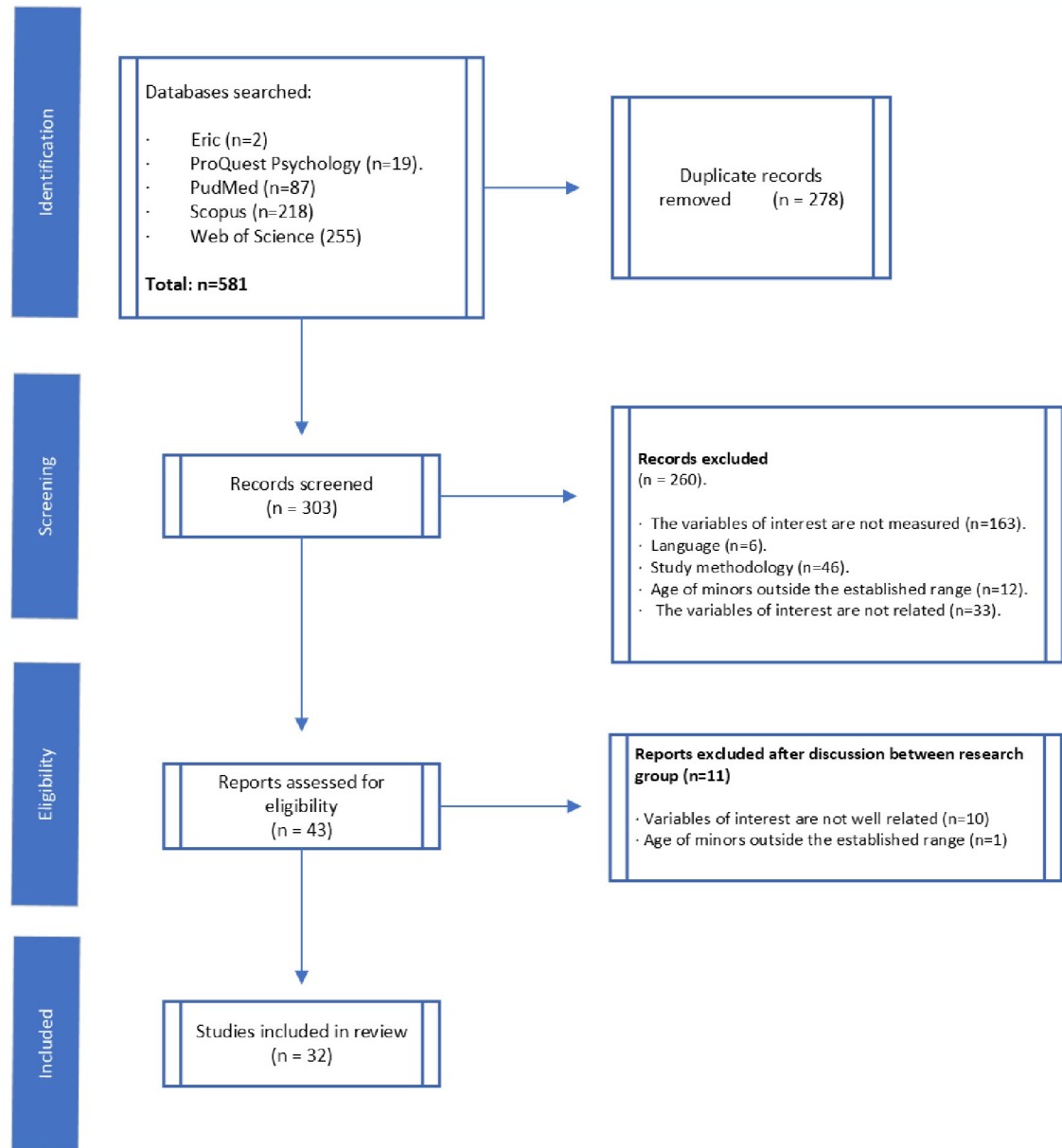

**Fig 1. Flowchart- Prism declaration of the study selection process.**

Kappa Index was 0,963. Only those articles that met all the inclusion criteria and none of the exclusion criteria were selected. For this purpose, a table was created that can be consulted in the following repository (https://bit.ly/repositorio_rs). Disagreements about the inclusion/ exclusion of the selected articles were resolved after consulting a third researcher and discussing their eligibility until reaching 100% agreement on the studies finally included. Third, to extract the results from the primary sources, both researchers extracted the results from each study independently to later pool them and check the concordance of the data for each of the studies. Furthermore, not all repositories are equally visible on the Internet or accessible to the researcher. In any case, we attempted to minimize bias by reviewing Google Scholar. Still, most publications on parental mediation and social media that emerged from peer-reviewed

channels did not meet all our inclusion criteria (i.e., language, relationship between variables, and compliance with minimum scientific research standards), and those that did, were contained in other reviewed databases. In addition, we analyzed a significant number of studies whose journals are not indexed in SCOPUS or WoS (mainly from India and Pakistan). Therefore, although there is a possibility of missing relevant studies, we took all steps to minimize it. Finally, for the synthesis and conclusions, it was decided to divide the results based on the different research objectives and results found, to facilitate their synthesis due to the complexity of the categories found. This work was carried out jointly by the two main researchers, with the rest of the researchers in charge of reviewing the concordance between the results, synthesis and conclusions. Although we are aware that it is very difficult to eliminate the risk of bias in this type of work, we consider that with the exhaustive procedure used, it has been possible to reduce it considerably.

## 2.4. Data extraction and qualitative analysis

Table 3 shows the details of the number of articles, the authors, the mediation strategies used, and the instruments used to measure PM in each article selected for this review.

Due to the number of conceptualizations of PM discussed above and to facilitate the data extraction to compare the results of the different studies, each work's PM was recategorized following the dichotomous proposal of Livingstone et al. [27]. Therefore, the results of the PM strategies will be presented in two macro-categories: enabling mediation (EM) and restrictive mediation (RM) strategies (see Tables 4 and 5). In studies in which other names were used for the PM strategies, the conceptual definition given by the authors and/or the semantic content of the items were considered to recategorize them into EM or RM. We note that, in the studies using several EM subcategories and presenting different or contrary results, a more detailed presentation of each result was made (see Tables 4 and 5).

Research on the use of SNs is extensive and complex. In the present review, the researchers analyzed the content of the articles to perform an orderly, coherent, and detailed analysis of the state of the art. To be able to answer each of the research questions, the following aspects of PM were analyzed: 1) usage time of SNs, 2) risks associated with SNs, and 3) benefits of SNs.

## 2.5. Quality assessment

Because majority of the included studies were cross-sectional, we used the Joanna Briggs Institute (JBI) Critical Appraisal Checklists for Analytical Cross-Sectional Studies to assess the risk of bias (methodological quality of studies) [55]. This is an eight-item checklist with four response options ("Yes", "No", "Unclear or "Not applicable"). Only "Yes" is scored with 1 point (while the others score 0 points). The total score ranges from 0 to 8. Although there are no explicit cut-off points, a common practice is to consider studies as high quality if they meet more than 80% of the criteria, moderate quality if they meet 50% to 80%, and low quality if they meet less than 50% [56].

## 3. Results

**3.1. Demographic, sociodemographic, and methodological characteristics of the analyzed studies.** Before delving into the relationship between PM and SNs, some general data on the studies carried out should be explored to provide a global and contextualized image of the state of the art.

Regarding the demographic characteristics of the final selection of articles (n = 32), the samples from 12 different countries were markedly heterogeneous. However, there was a predominance of research with samples from the United States (n = 8), China and Spain (n = 3),

**Table 3. Synthesis of sociodemographic and methodological data of the reviewed studies.**

| First author/ Year of publication | Country of the sample | SS | Sex (% women) Sex ratio B: G | Sample age (M and SD) | Sample of parents/ guardians (M and SD) | D | DPM | Type of mediation (depending on primary source author) | Mediation categorization for review | Parental Mediation Instruments | Quality asessment |
|---|---|---|---|---|---|---|---|---|---|---|---|
| Albeladi, N. (2020) | Saudi Arabia | 393 | 68.70% [46:100] | 15.61 (SD = 1.66) | | CS | Yes | • Active Mediation of Internet Use. • Active Mediation of Internet Security • Restrictive Mediation • Technical Mediation • Monitoring | Enabling & Restrictive | AQ | H |
| Álvarez-García, D. (2018) | Spain | 3059 | 48.50% [106:100] | 14.01 (SD = 1.39) | | CS | No | • Restrictive Mediation • Supervision | Enabling & Restrictive | CQV | M |
| Barry, C.T (2023) | United States | 316 | 52.5% [90:100] | 15.64 (SD = 1.09) | 45,33 (SD = 8.36) | CS | Yes | • Parental Control. • Parental Communication | Enabling & Restrictive | AHI | H |
| Charmaraman, L. (2022) | United States | 773 | 49.70% [101:100] | 12.6 (SD = 0.96) | | CS | No | • Restrictive Mediation | Restrictive | AHI | H |
| Chen, H. (2016) | United States | 622 | 51.13% [96:100] | 14.94 (SD = 1.60) | 47.08 (SD = 8.60) | CS | Yes | • Restrictive Mediation • Instructive Mediation. | Enabling & Restrictive | AHI | H |
| Chen, H. (2023) | China | 642 | 63.10% [58:100] | 16.09 (SD = 1.51) | 44.49 (SD = 7.91) | CS | Yes | • Active Mediation • Restrictive Mediation • Non-intrusive inspection | Enabling & Restrictive | AQ | H |
| Chou, H. (2019) | Taiwan | 655 | 54.00% [85:100] | - | | CS | No | • Restrictive Mediation • Instructive Mediation. | Enabling & Restrictive | CQV | M |
| Corcoran, E. (2022) | United States | 1021 | 41.50% [141:100] | 12.12 (SD = 1.37) | 38.18 (SD = 7.44) | CS | No | • Education Mediation • Talk Mediation • Co-use • House Rules | Enabling & Restrictive | AHI | H |
| Dhir (2019) | - | 1552 | 48.70% [105.34:100] | 14.42 (SD = 1.05) | | CS | Yes | • Parental Encouragement • Parental Worry • Parental Monitoring • Parental Permission | Enabling & Restrictive | AQ | H |
| Fardouly, J. (2022) | Australia | 498 | 48% [108.33:100] | 12.59 (SD = 0.52) | 45.29 (SD = 4.28) | LO | No | • Parental Control | Restrictive | AHI | H |
| Fardouly, J. (2018) | Australia | 284 | 53.20% [88:100] | 11.2 (SD = 0.56) | | CS | Yes | • Parental Control | Restrictive | AHI | H |

*(Continued)*

**Table 3.** (*Continued*)

| First author/ Year of publication | Country of the sample | SS | Sex (% women) Sex ratio B: G | Sample age (M and SD) | Sample of parents/ guardians (M and SD) | D | DPM | Type of mediation (depending on primary source author) | Mediation categorization for review | Parental Mediation Instruments | Quality asessment |
|---|---|---|---|---|---|---|---|---|---|---|---|
| Hampton K.N. (2023) | United States | 3258 | 52.30% [91.20:100] | - | | CS | Yes | • Restrictive Mediation<br>• Instructive Mediation | Enabling & Restrictive | AHI | H |
| Hol, S. (2017) | Singapore | 1424 | 47.60% [110:100] | - (SD = 1.88) | | CS | Yes | • Active Mediation<br>• Restrictive Mediation. | Enabling & Restrictive | AQ | H |
| Kang, H. (2022) | China | 500 | 46.20% [116:100] | 15.5 (SD = 1.72) | | CS | Yes | • Active Mediation<br>• Restrictive Mediation. | Enabling & Restrictive | AQ | H |
| Koning (2018) | The Netherlands | 352 | 51% [96:100] | 13.9 (SD = 0.74) | | LO | Yes | • Internet-specific rules<br>• Reactive rules<br>• Frequency of communication<br>• Quality of communication | Enabling & Restrictive | AQ | H |
| Lee, N. (2021) | South Korea | 184 | 55% [82:100] | 11.34 (SD = —) | | CS | Yes | • Active mediation of Internet use<br>• Active mediation of Internet security<br>• Monitoring<br>• Technical mediation<br>• Restrictive mediation | Enabling & Restrictive | CQV | H |
| Len-Ríos, M. (2016) | United States | 354 | 50% [100:100] | 13.21 (SD = —) | | CS | No | • Monitoring | Restrictive | AQ | M |
| Lui, C. (2016) | Singapore | 780 | 50.90% [96:100] | 13.94 (SD = 0.90) | | CS | Yes | • Active Mediation<br>• Restrictive Mediation | Enabling & Restrictive | AQ | H |
| Lui, C. (2013) | Singapore | 780 | - | 13.94 (SD = 0.90) | | CS | Yes | • Active Mediation<br>• Restrictive Mediation | Enabling & Restrictive | AQ | H |
| Lui, C. (2019) | - | 351 | 50.40% [98:100] | 13.98 (SD = 0.94) | | CS | Yes | • Active Mediation<br>• Restrictive Mediation | Enabling & Restrictive | AQ | H |
| López- De- Ayala, M.C. (2021) | Spain | 517 | 48.60% [106:100] | 13.53 (SD = 1.19) | | CS | Yes | • Parental Well-being Mediation<br>• Co-viewing<br>• Restrictive Mediation<br>• Mediation requested by children. | Enabling & Restrictive | AQ | H |

(*Continued*)

**Table 3.** (Continued)

| First author/ Year of publication | Country of the sample | SS | Sex (% women) Sex ratio B: G | Sample age (M and SD) | Sample of parents/ guardians (M and SD) | D | DPM | Type of mediation (depending on primary source author) | Mediation categorization for review | Parental Mediation Instruments | Quality asessment |
|---|---|---|---|---|---|---|---|---|---|---|---|
| Martín-Criado, J.M. (2021) | Spain | 6182 | 49% [104:100] | 12.60 (SD = 1.65) | | CS | Yes | • Parental Supervision | Enabling | D/S | H |
| Martins, N. (2019) | United States | 655 | 54% [85:100] | 15.59 (SD = 1.12) | | CS | No | • Active mediation (Authoritative, Controlling, and Inconsistent) <br>• Restrictive mediation (Authoritative, Controlling, and Inconsistent) | Enabling & Restrictive | AQ | H |
| Mesch, G. (2018) | United States | 462 | 49% [104.100] | 14.51 (SD = —) | 45.06 (SD = 10.37) | CS | Yes | • Parental Control | Restrictive | AHI | H |
| Rudnova, N. (2023) | Russia | 4011 | 58% [72:100] | 14.07 (SD = 0.76) | | CS | Yes | • Parental Support <br>• Parental Control | Enabling & Restrictive | CQV | H |
| Shin, W. (2014) | Malaysia. | 469 | 54.60% [83:100] | 13.5 (SD = 0.50) | | CS | Yes | • Active Mediation <br>• Restrictive Mediation | Enabling & Restrictive | AHI | H |
| Sun, X. (2021) | China | 823 | 49.94% [100:100] | 12.41 (SD = 1.23) | | CS | Yes | • Active parental mediation | Enabling | AQ | H |
| Symons, K. (2020) | Belgium | 357 | 54.90% [82:100] | 15.73 (SD = 1.50) | Mothers: 44.19 (SD = 4.72). Fathers: 46.67 (SD = 5.65) | CS | Yes | • Open parent-child communication about Internet use | Enabling | AQ | H |
| Symons, K. (2020) | Belgium | 357 | 54.90% [82:100] | 15.73 (SD = 1.50) | Mothers: 44.19 (SD = 4.72). Fathers: (46.67 (SD = 5.65) | CS | YES | • Parental control | Restrictive | AQ | H |
| Vaala, S.E. (2015) | United States | 629 | 48.40% [107:100] | 14.5 (SD = —) | 44.4 (SD = —) | CS | Yes | • Restrictive mediation <br>• Monitoring <br>• Co-viewing | Enabling & Restrictive | AQ | H |
| Wright, M. (2018) | United States | 567 | 52% [92:100] | 13.48 (SD = —) | | LO | YES | • Instructive Mediation <br>• Co-viewing <br>• Restrictive Mediation | Enabling & Restrictive | AQ | H |
| Yépez-Tito, P. (2020) | Ecuador | 613 | 44.04% [127:100] | 14.62 (SD = 1.71) | | CS | No | • Parental Supervision. | Restrictive | CQV | H |

*Note.* SS = Sample Size; Sex ratio: number of boys per each hundred girls, first decimal has been rounded; B = Boys; G = Girls; M = Mean; SD = Standard deviation; D = Design; CS = Cross-sectional, LO = Longitudinal; DPM = Defines Parental Mediation;; AQ = Adaptation of a questionnaire; CQV = Complete Questionnaire Validated; AHI = Ad hoc items; D/S = Dimension or Scale of a questionnaire; Quality represents literature quality evaluation; H = high quality, MD = medium quality.

followed by Singapore and Australia (n = 2), and Saudi Arabia, Belgium, Russia, Taiwan, South Korea, the Netherlands, Ecuador, and Malaysia (n = 1). Two studies did not specifically indicate the sample's origin (see Table 3). The total sample consisted of 32303 adolescents (the sample of two articles was not counted in the sum because it had been used in other previous

**Table 4. Main results obtained by the authors of the relationship between parental mediation and time spent on social networks.**

| Category | Principal author and year of publication | Type of mediation | Results from primary sources | Categorization of results for systematic review |
|---|---|---|---|---|
| USAGE TIME | Albeladi, N. (2020) | R: Restrictive mediation (RM) | RM = Frequency of Snapchat and Instagram usage | R = NR |
| | | E: Active mediation of internet use (AMIU) | AMIU = Frequency of use. | E = CR |
| | | Active mediation of Internet safety (AMIS) | AMIS = Frequency of use | |
| | | Technical Mediation (TM) | ↑ TM ↓ Frequency of Snapchat and Instagram usage | |
| | | Monitoring (M) | ↑ M ↑ Frequency of use of Snapchat and Instagram | |
| | Álvarez-García, D. (2018) | R: Restrictive Mediation (RM) | ↑ RM ↓ Usage time | ↑ R ↓ Usage time |
| | | E: Parental Supervision (PS) | ↑ PS↓ Usage time | ↑ E ↓ Usage time |
| | Barry, C.T (2023) | R: Parental Control (PC) | ↑ PC.↓ Use of SNs | ↑ R ↓ Usage time |
| | | E: Open Communication (OC) | ↑ OC ↑ Use of SNs | ↑ E ↑ Usage time |
| | Chen, H. (2016) | R: Restrictive Mediation (RM) | RM. = Use of SNs | R = NR |
| | | E: Instructive Mediation (IM) | ↑ I.M. ↑ Use of SNs | ↑ E ↑ Usage time |
| | Fardouly, J. (2018) | R: Parental Control. (PC) | ↑ PC ↓ Usage time. | ↑ PC ↓ Usage time |
| | | | ↑ R ↓ Usage time | ↑ R ↓ Usage time |
| | Fardouly, J. (2022) | R: Parental Control. (PC) | ↑ PC ↓ Usage time. | ↑ R ↓ Usage time |
| | Hampton K.N. (2023) | R: Restrictive Mediation (RM) | ↑ RM ↓ Usage time (Only in girl) | ↑ R ↓ Usage time (Only in girl) |
| | | E: Instructive Mediation (IM) | ↑ IM ↓ Usage time | ↑ E ↓ Usage time |
| | Kang, H. (2022) | R: Restrictive Mediation (RM) | ↑ RM ↑ Usage time in Douyin | ↑ R ↑ Usage time |
| | | E: Active Mediation (AM) | ↑ AM ↑ Usage time in Douyin | ↑ E ↑ Usage time |
| | Lee, N. (2021) | R: Restrictive mediation (RM) | RM = Usage time. | R = NR |
| | | E: Active mediation of internet use (AMIU) | AMIU = Usage time | ↑ E ↓ Usage time. |
| | | Active mediation of Internet safely (AMIS) | AMIS = Usage time | |
| | | Technical Mediation (TM) | ↑ TM ↓ Usage time | |
| | | Monitoring (M) | M = Usage time | |
| | Len-Ríos, M. (2016) | R: Monitoring (M) | M. = Usage time | R = NR |
| | López- De- Ayala, M.C. (2021) | R: Restrictive Mediation (RM) | ↑ RM ↓ Usage time | ↑ R ↓ Usage time |
| | | E: Well-being Mediation (WM) | WM = Usage time | E = NR |
| | | Monitoring (M) | M = Usage time | |
| | | Mediation requested by the children (MRC) | MRC = Usage time | |
| | Martín-Criado, J.M. (2021) | E: Parental Supervision (PS) | ↑ PS ↓ Usage time | ↑ E ↓ Usage time |
| | Martins, N. (2019) | R: Controlling Restrictive (CR) | CR = Usage time | ↑ R ↑ Usage time |
| | | Autonomy Supportive Restrictive (ASR) | ASR = Usage time | ↑ E ↑ Usage time |
| | | Inconsistent Restrictive (IR.) | ↑ IR ↑ Usage time | |
| | | E: Autonomy Supportive Active (ASA) | ASA = Usage time | |
| | | Controlling Active (CA) | ↑ CA ↑ Usage time | |
| | | Inconsistent Active (IA) | ↑ IA ↑ Usage time | |
| | Mesch, G. (2018) | R: Parental Control (PC) | ↑ PC ↓ Activity in SN | ↑ R ↓ Usage time |
| | Rudnova, N. (2023) | R: Parental Control (PC) | ↑ PC ↑ Usage time | ↑ R ↑ Usage time |
| | | E: Parental Support (PS) | ↑ PS ↓ Usage Time | ↑ E ↓ Usage time |
| | Vaala, S.E. (2015) | R: Restrictive Mediation (RM) | RM = Usage time | R = NR |
| | | E: Parental Internet Tracking. (PIT.) | PIT = Usage time | E = NR |
| | | Co-viewing (CV) | CV = Usage time. | |
| | Yépez-Tito, P. (2020) | R: Parental Supervision (PS) | ↑ PS ↑ Usage time on Snapchat and Instagram | ↑ R ↑ Usage time on Snapchat and Instagram |

*Note.* CR = Contradictory Results; E = Enabling; NR = No Relations; R = Restrictive

**Table 5. Main results obtained by the authors of the relationship between parental mediation and main risks in social networks.**

| Category | Principal author and year of publication | Type of mediation | Results from primary sources | Categorization of results for systematic review |
|---|---|---|---|---|
| RELATIONAL RISKS | Álvarez-García, D. (2018) | R: Restrictive Mediation (RM) | ↑ RM ↓ Probability of performing as a cyberaggressor | ↑ R ↓ cyberaggressor |
| | | E: Parental Supervision (PS) | ↑ PS ↓ Probability of performing as a cyberaggressor | ↑ E ↓ Cyberaggressor |
| | Charmaraman, L. (2022) | R: Restrictive Mediation (RM) | ↑ RM ↑ Contacts in children | ↑ R ↑ Contacts in children |
| | | | ↑ RM ↓ Contacts in adolescents | ↑ R ↓ Contacts in adolescents |
| | Chen, H. (2023) | R: Restrictive Mediation (RM) | RM = Probability of performing as a cyberaggressor | R = NR Probability of performing as a cyberaggressor. |
| | | E: Active Mediation | RM = Probability of being a cybervictim | R = NR Cybervictim. |
| | | | ↑ AM = ↓ Probability of performing as a cyberaggressor | E ↓ Cyberaggressor. |
| | | | ↑ AM = ↓ Probability of being a cybervictim | E ↓ Cybervictim. |
| | Ho S. (2017) | R: Restrictive Mediation (RM) | ↑ RM ↓ Probability of performing as a cyberaggressor | ↑ R ↓ Cyberaggressor |
| | | E: Active Mediation (AM) | ↑ AM ↓ Probability of performing as a cyberaggressor | ↑ E ↓ Cyberaggressor |
| | Martín-Criado, J.M. (2021) | E: Parental Supervision (PS) | ↑ PS ↓ Probability of being a cybervictim | ↑ E ↓ Cybervictim. |
| | Mesch, G. (2018) | R: Parental Control (PC) | PC = Probability of being a cybervictim | R = NR Cybervictim |
| | Shin, W. (2014) | R: Restrictive Mediation (RM) | ↑ RM ↑ Contacts with strangers | ↑ R ↑ Contacts with strangers |
| | | E: Active Mediation (AM) | ↑ AM ↓ Contacts with strangers | ↑ E ↓ Contacts with strangers |
| | Symons, K. (2020) | E: Open parent-child communication (OPCC) | ↑ OPCC ↓ Contacts with strangers | ↑ E ↓ Contacts with strangers |
| | Symons, K. (2020) | R: Parental Control (PC) | ↑ PC ↓ Contacts with strangers | ↑ R ↓ Contacts with strangers |
| | Wright, M. (2018) | R: Restrictive Mediation (RM) | ↑ RM ↑ Probability of being a cybervictim | ↑ R ↑ Cybervictim |
| | | E: Instructive Mediation (IM) | ↑ IM ↓ Probability of being a cybervictim | ↑ E ↓ Cybervictim |
| | | Co-viewing (CV) | ↑ CV ↓ Probability of being a cybervictim. | |
| PRIVACY RISKS | Chen, H. (2016) | R: Restrictive Mediation (RM) | ↑ RM ↓ Disclosure of personal information | ↑R ↓ Disclosure |
| | | E: Instructive Mediation (IM) | IM = Disclosure of personal information | E = NR Disclosure. |
| | | | RM = Positive contact management | R = NR Positive management. |
| | | | ↑ I.M. ↑ Positive contact management | ↑ E ↑ Positive management |
| | Chou, H. (2019) | R: Restrictive Mediation (RM) | ↑ R.M. ↑ Positive management of privacy on Facebook | ↑ R ↑ Positive management |
| | | E: Instructive Mediation (IM) | ↑ IM ↑ Positive management of privacy on Facebook | ↑ E ↑ Positive management |
| | Corcoran, E. (2022) | R: House Rules (HS) | HS = Privacy Concerns | R = NR Privacy Concerns |
| | | E: Talk (T) | ↑ T ↓ Privacy Concerns | E = CR |
| | | Education (E) | ↑ E ↑ Privacy Concerns | |
| | | Co-use (CU) | ↑ CU ↓ Privacy Concerns | |
| | Kang, H. (2022) | R: Restrictive Mediation (RM) | RM = Disclosure of personal information | R = NR Disclosure |
| | | E: Active Mediation (AM) | ↑ RM ↑ Limits of sharing information | ↑ R ↑ Limits sharing |
| | | | ↑ RM ↑ Management control | ↑ R ↑ Control |
| | | | ↑ AM ↑ Disclosure of personal information | ↑ E ↑ Disclosure |
| | | | ↑ AM ↑ Limits of sharing information | ↑ E ↑ Limits sharing |
| | | | AM = Management control | E = NR Control |
| | Liu C. (2016) | R: Restrictive Mediation (RM) | ↑ RM & AM ↓ Disclosure of personal textual information on Facebook | ↑ R & E ↓ Text disclosure |
| | | E: Active Mediation (AM) | ↑ RM & AM ↓ Disclosure of personal images on Facebook | ↑ R & E ↓ Disclosure images |

(*Continued*)

**Table 5.** (Continued)

| Category | Principal author and year of publication | Type of mediation | Results from primary sources | Categorization of results for systematic review |
|---|---|---|---|---|
| | Liu C. (2013) | R: Restrictive Mediation (RM) | ↑ RM ↑ Privacy Concerns | ↑ R ↑ Privacy Concerns |
| | | E: Active Mediation (AM) | ↑ AM ↑ Privacy Concern | ↑ E ↑ Privacy Concerns |
| | | | ↑ RM ↓ Disclosure of Personal Information | ↑ R ↓ Disclosure |
| | | | ↑ AM ↓ Disclosure of Personal Information | ↑ E ↓ Disclosure |
| | Liu, C. (2019) | R: Restrictive Mediation (RM) | ↑ RM & AM ↑ Concern about Facebook privacy | ↑ R and H ↑ Concerns |
| | | E: Active Mediation (AM) | RM & AM = Facebook Information Disclosure | R and H = NR Disclosure |
| | Mesch, G. (2018) | R: Parental Control (PC) | ↑ PC ↓ Having a public profile | ↑ R ↓ Public profile |
| | | | ↑ PC ↓ Lying about age | ↑ R ↓ Lying about age |
| | | | ↑ PC ↓ Sharing passwords | ↑ R ↓ Sharing passwords |
| | Shin, W. (2014) | R: Restrictive Mediation (RM) | ↑ RM ↑ Disclosure of private information | ↑ R ↑ Disclosure |
| | | E: Active Mediation (AM) | ↑ AM ↑ Disclosure of private information | ↑ E ↑ Disclosure |
| HEALTH RISKS | Albeladi, N. (2020) | R: Restrictive mediation (RM) | RM. = Addiction to SNs | R = NR Addiction to SNs |
| | | E: Active mediation of internet use (AMIU) | AMIU = Addiction to SNs | E = NR Addiction to SNs |
| | | Active mediation of Internet safely (AMIS) | AMIS = Addiction to SNs | |
| | | Technical Mediation (TM) | TM = Addiction to SNs | |
| | | Monitoring (M) | M = Addiction to SNs | |
| | Charmaraman, L. (2022) | R: Restrictive Mediation (RM) | ↑ RM ↓ FoMO children | ↑ R ↓ FoMO children |
| | | | ↑ RM ↑ FoMO adolescents | ↑ R ↑ FoMO adolescents |
| | Dhir (2019) | R: Parental Permission (PP) | ↑ PP ↓ Social Media Fatigue | ↑ R = CR |
| | | Parental Worry (PW) | ↑ PW ↑ Social Media Fatigue | ↑ E ↑ Social Media Fatigue |
| | | E: Parental Encouragement (PE) | ↑ PE ↑ Social Media Fatigue | |
| | | Monitoring (M) | M = Fatigue | |
| | Koning (2018) | R: Reactive Rules (Reac.R.) | ↑ Reac.R. ↑ Social Media Disorder | R = CR |
| | | Restrictive Rules (Res.R.) | ↑ Res. R. ↓ Social Media Disorder | ↑ E ↓ Social Media Disorder |
| | | E: Frequency Communication (FC) | FC = Social Media Disorder | |
| | | Quality communication (QC) | ↑ QC ↓ Social Media Disorder | |
| | Rudnova, N. (2023) | R: Parental Control (PC) E: Parental Support (PS) | ↑ PC ↓ Social Media Addiction ↑ PS ↑ Social Media Addiction | ↑ R ↓ Social Media Addiction ↑ E ↑ Social Media Addiction |
| | Sun, X. (2021) | E: Active Mediation (A.M.) | ↑ A.M. ↓ Addiction to SNs | ↑ E ↓ Addiction to SNs |

*Note.* CR = Contradictory Results; E = Enabling; NR = No Relations; R = Restrictive

articles also included in this study), with an age range between 9 and 18 and a mean age of 13.93 years. In six of these studies, there was also a sample of parents and legal guardians, consisting of 2925 participants, aged between 31 and 70 years and a mean age of 44.47 (see Table 3). There was a greater predominance of studies with a higher percentage of girls (n = 17) than boys (n = 13); one study had the same number of participants of both sexes, and another did not specify the participants' sex. Concerning the sample size of the different papers, 13 papers had up to 500 participants, 12 papers presented samples of 500 to 1000 participants, 3 papers used samples of 1000 to 2000 participants, and the remaining 4 papers had samples of more than 2000 participants. As for the quality assessment, 90,6% of the studies presented high quality, and 9,4% medium quality, no studies were of low quality (see Table 3).

Regarding the methodological characteristics of the studies, all of them was observational studies. Only three of them were longitudinal studies, whereas the rest were cross-sectional (n = 29). Concerning PM assessment, most studies used an adaptation of a previously validated

questionnaire (n = 17), followed by items created ad hoc (n = 9), or they created a new questionnaire and validated it (n = 5), and one study used one dimension of a previously validated questionnaire (n = 1). We note the high variability of taxonomies and nomenclatures (a total of 27 different nomenclatures of the PM strategies) found in the PM construct; whereas 11 studies considered a single dimension of PM, 13 were two-dimensional, and the rest (n = 9) used 3 or more dimensions to evaluate the construct (see Table 3). In addition, the PM variable was only defined in 20 of the analyzed studies.

## 3.2. Relationship between PM strategies and the use of SNs

The relationships between PM strategies and the usage time of SNs, as well as the risks and benefits of SNs, are presented sequentially.

**3.2.1. PM and the children's usage time of SNs.** We selected the works relating PM strategies to the youngsters' SN usage time and frequency to answer this research question. A total of 16 studies were found that linked the two variables (see Table 4).

Regarding RM strategies, five of the studies found no relationship between young people's SN usage time and frequency and RM strategies [57–61]. In seven studies, RM was related to children's shorter SN usage time [19, 21, 62–66], and in one study, this relationship was found only in girls [65]. In contrast, in four studies, this type of mediation was associated with a greater SN usage time and frequency [67–70] (see Table 4).

On the other hand, regarding EM strategies, five of the studies found an association with a lower SN usage time and frequency [59, 62, 65, 69, 71]. In four of the works, EM was associated with youngsters' longer SN usage time and frequency [58, 63, 67, 68]. Moreover, one article found contradictory results between the different EM strategies [57], and in other two papers no relationship between the two variables were found [61, 66] (see Table 4).

**3.2.2. Relationship between PM and risks in SNs.** Below are the results of the studies relating PM to different risks to which children are exposed in SNs (see Table 5).

After analyzing the content of the 32 selected articles, three categories were created to order the risks in SNs according to their thematic nature: relational risks (derived from the interaction between people on the SNs), privacy risks (related to the disclosure of personal information and privacy), and health risks (addiction to SNs, Social Media Fatigue, Fear of Missing Out—FoMO—, and Social Media Disorder.).

• Relationship between PM and relational risks.

Firstly, in this regard, 10 studies addressed the relationship between PM and cyberaggression, cybervictimization, and contact with strangers. From these studies, it can be deduced that RM strategies have been associated with lower scores of cyberaggression [42, 62] and contact with strangers [72]. On the contrary, a relationship was also found between the RM strategies and an increase in young people's cybervictimization [73] and contact with strangers [74]. Another two studies found no relationship between RM strategies and the likelihood of being a cyberaggressor [75] or a cybervictim [21, 75]. Concerning EM strategies, they seem to be related to a lower probability of being a cyberbully [42, 62, 75], a cybervictim [71, 73, 75], or having contact with strangers [74, 76].

• Relationship between PM and privacy risks.

Secondly, regarding the relationship between PM and privacy risks, nine works were found relating these variables. Thus, on the one hand, in three articles, RM was associated with lower disclosure of information in SNs [43, 58, 67]. It was also shown to protect against other privacy-related risk behaviors, such as the probability of sharing passwords or having a public

profile [21]. In other studies, RM was associated with increased privacy concerns [77] and better privacy management practices [78]. In only one study was RM associated with an increased likelihood of disclosing private information [74], whereas another two works found no relationship between RM and privacy concerns and disclosure of private information [67, 79]. On the other hand, in one article, EM was related to positive management of contacts in SNs [58] and, in another, to better privacy practices [78]. However, in other investigations, EM was related to higher privacy disclosure scores on SNs [67, 74] and, in one work, to sharing information on SN [67]. Another study found contradictory results between different PM strategies [79]. In addition, in two articles, PM was not divided into dichotomous typologies, but was addressed globally. In these works, PM was found to be a protective factor against the disclosure of private information [80] and for youngsters' greater concern about their privacy [81].

- Relationship between PM and health-related risks in SNs.

Thirdly, six studies addressed the relationship between PM strategies and SN risks related to youth's health. It was found that RM is related to lower rates of Social Media Fatigue (18), FoMO in childhood [16], and addiction to SNs [69]. In contrast, two studies were linked to greater mental health problems, specifically Social Media Disorder [17] and FoMO in adolescence and preadolescence [16]. One study found no link between RM and health [57]. The results are mixed regarding EM. In one study, EM was associated with lower addiction to SNs [82] and Social Media Disorder [17]. Conversely, in another two studies, EM was linked to increased Social Media Fatigue [18] and addiction to SNs [69].

**3.2.3. Relationship between PM and the benefits of SNs.**   In this regard, none of the 32 articles analyzed reported information on the relationship between the different PM strategies and the benefits of SNs.

## 4. Discussion

Everything that concerns the use of SNs in adolescence seems to be a complex line of research, especially when the figure of the parents intervenes through PM strategies. In this sense, previous studies that measured PM dichotomously reported that the existence of PM (regardless of the type and degree) is preferable to its absence [26, 83, 84]. In this systematic review, we intended to go a step further, providing systematized knowledge about the possible evidence of the use of SNs and its relationship with the different PM strategies. Specifically, we focused on analyzing the relationship between families' PM and their children's usage time of SNs, the effects that different PM strategies have on reducing the risks of SNs, and how different PM strategies are relate to the benefits associated with SNs. For this purpose, a broad definition like that by Livingstone et al. [27] was used to define PM as the set of efforts by the family to maximize the benefits and reduce the risks of the Internet; as well as presenting two basic categories of mediation with empirical evidence: enabling and restrictive mediation.

In order to answer the first research question of this review, we note that one of the parents' main concerns is the amount of time young people spend on SNs [85]. In this sense, the results suggest that PM strategies are associated with a shorter usage time and frequency of SNs. However, we found more evidence in favor of RM [19, 21, 62, 64, 86, 87] than EM [21, 59, 62]. These data in favor of RM as a more efficient strategy in reducing the usage time of SNs align with those reported by a meta-analysis of digital media [39]. This study also points out that RM is associated with shorter usage time, whereas active mediation (EM) does not affect usage time. One of the possible explanations for this may be related to the fact that there are more articles measuring exclusively RM [19, 21, 60, 64, 70] than those measuring EM [71]. Another possible explanation is that the parents' main objective is for their children to reduce the

amount of time they spend on SNs and not so much for the children to use SNs autonomously and responsibly. For this purpose, RM strategies are a faster, more effective, and therefore more available way for parents to reduce usage time and frequency [41]. A fact to highlight within this category is that technical mediation (framed within EM) achieves better results than the rest of the EM strategies [57, 59], which could be of interest for the role of technical mediation in the face of these problems. In these studies, technical mediation aligns with the data obtained by RM (see Table 4). This could be because this strategy is conceptually closer to RM than to EM, despite the results found by Livingstone et al. [27]. In any case, there is no clear directionality between PM strategies and SN usage time.

In relation to the second research question, one of the worrisome aspects that the use of SNs has generated in society and academia has been the set of relational and dysfunctional risks associated with their use [15]. To date, the most researched risk has been cyberbullying [21, 42, 62, 71, 73]; EM has a greater relationship with the reduction of this problem than RM. A possible explanation is that cyberbullying is a more relational risk and has multiple dimensions related to different Internet usage profiles [62, 88], so it cannot be addressed simply with restrictive strategies. In addition, EM is related to better communication and, therefore, is a protective factor against cybervictimization [88]. However, some studies show that RM is effective [42, 62], although it may also produce the opposite effect [73]. Taken together, these results suggest some consensus about the relevance of a specific PM strategy compared to another to minimize cyberbullying. However, in line with other reviews [89], more research and further clarification of the relationship between PM and cyberbullying are needed.

Minors' contact with adult strangers through SNs is another of the most studied relational risks [16, 72–74, 76]. The results are very similar to those found for cyberbullying (see Table 5). A possible explanation for this parallelism may be the shared characteristics of these Internet risks, as they are both relational risks [15]. However, it is noteworthy that one study relates a greater presence of RM with more contact with strangers [16, 74]. This could be related to the "boomerang" effect that this type of restriction can produce in youngsters and adolescents, making what was intended to be forbidden more desirable [21, 90]. This aspect may be of interest for recommendations about parenting styles.

The last of the risks analyzed is related to the privacy disclosure on SNs. Thus, the results indicate that both mediation strategies (RM and EM) can be considered effective for reducing the disclosure of personal information and improving privacy protection. However, RM was more closely associated with reducing behaviors that reveal private information than EM. These results are consistent with the systematic review of [45], where RM practices were more effective in regulating children's privacy behaviors in the online context. A possible explanation for these results is that RM strategies involve the prohibition or establishment of rules for using digital media (e.g., limitations when using the computer camera or restrictions when posting photos or videos on SNs). This can be a protective factor for safeguarding the child's privacy [91]. These findings provide a complete answer to the second research question of this study.

The last of the risks analyzed is related to the use of SNs and their link to different mental health problems, where PM strategies showed various degrees of effectiveness. RM is associated with a lower risk of FoMO in childhood and a higher risk in adolescence [16]. A possible explanation for this is that adolescents use SNs much more than children, and this is related to higher levels of FoMO [16]. Another reason could be the lack (or lower frequency) of PM in adolescence, because as the children grow older, PM decreases [92]. Likewise, EM is related to a lower score in addition to SNs and lower Social Media Disorder [17, 57, 82]. In contrast, EM is associated with higher scores in Social Media Fatigue [18]. A possible interpretation of these data is that, despite being aspects related to mental health, they are different phenomena. On

the one hand, addiction to SNs and Social Media Disorder are problems with a clinical-diagnostic approach [17, 57, 82], whereas Social Media Fatigue presents a more global (and less clinical) approach [18]. However, we must conclude that the results obtained do not clearly point in a single direction. Hence, the conclusions align with other reviews on the problematic use of the Internet or online gambling, in which no kind of PM was consistently associated with adolescents' higher or lower rates of problematic screen use [40].

Finally, no results were obtained about the third research question, which sought to analyze the relationship between PM and the benefits of SNs. Neither of the two variables was related in any of the studies analyzed. This is a significant finding because, according to the review literature, SNs have a number of benefits, such as reducing isolation, improving social skills, providing a platform for continuous communication [12, 93], and helping young people with mental health problems [12]. It is paradoxical that, although the definition of PM generally includes increased benefits associated with new technologies [27], no study has taken this factor into account when investigating its relationship with SNs. However, it indicates a necessary study path to fully understand the two axes of PM: preventing risks–maximizing benefits.

After all that has been said, it is important to interpret these results cautiously because despite our findings, it is impossible to draw categorical conclusions about the appropriateness of one type of mediation over another in terms of the uses and risks of SNs. However, it could be said that an effort has been made to highlight the common patterns concerning PM and some uses and risks of the Internet. This work can be added to those by other authors in this line of research [39, 40, 45, 89, 94].

This study also has some limitations. First, this review focused on peer-reviewed scientific articles. In this sense, we considered using other less restrictive databases such as Google Schoolar, but we discarded this idea because we could not use the same algorithm as in the rest of the databases, and also due to the large volume of works found. Therefore, the search was limited to specific scientific databases. Secondly, the use of a two-dimensional category in PM strategies allows a parsimonious analysis of the results and a global and systematic approach to the state of the art, which besides, has an empirical basis [27] but may also detract from the specificity of the content. Thirdly, it should be noted that the analysis did not consider the possible differences in the stages of adolescence (because the range of 9–18 is very broad). Fourthly, in cases where it was unclear which type of PM was used, the authors proposed a categorization based on its definition and on that provided by Livingstone et al. [27], which may have led to some problems of interpretation. Finally, only studies in English and Spanish were included in this review, so papers of interest in other languages may have been excluded.

Now, we present some future lines of research. In line with the statements of Garmendia et al., 60% of parents say they need training on Internet use and online security, so it is necessary to design PM training programs [95]. This training should consider the different existing mediation strategies in a complementary way and not as mutually exclusive mediation styles from a meta-cognitive level. On the other hand, as has been shown [36, 37, 94], we need a greater methodological and conceptual specificity about PM to be able to compare the results better and carry out meta-analyses. In this context, it might be beneficial to initially explore international projects related to this topic or widely used instruments/taxonomies within the field. Likewise, it should be determined whether those measurement instruments of PM have a second-order factor that can evaluate EM and RM strategies as parts of a whole, that is, a global strategy of PM. In addition, it would be appropriate to change the focus of many studies asking about the most effective mediation strategy rather than what combination of strategies could be more effective (as this problem is complex and multifaceted), establishing "profiles" when applying PM strategies. Furthermore, it is essential for research to examine how PM may (or may not) favor minors' benefits from SNs. This is a fundamental aspect when it comes to

promoting the relationship of young people with SNs, and the absence of research is remarkable. Another aspect to consider is the relationship between PM strategies and other closely related variables, such as traditional parenting styles, which could modulate the effect of PM on the associated benefits and risks. Moreover, there has also been a lack of evidence linking the perceptions of children and parents about the application of PM strategies. Finally, more longitudinal studies are needed to understand the transition of PM strategies and risk behaviors in SNs throughout the different developmental and educational stages.

The conclusions of this study indicate that, in general, PM strategies are a useful resource for reducing the usage time and the risks associated with SNs. However, there are no results about their relationship with the benefits of SNs. Thus, it could be inferred from the evidence found that when it comes to reducing behaviors in which the reduction of risk behaviors takes precedence (for example, limiting time, use of programs, frequency or protecting privacy) RM is more effective. However, when promoting the child's or adolescent's autonomy and responsible use, then EM seems more effective in preventing relational risks such as contact with unknown adults and mental health issues such as addiction to SNs or Social Media Disorder. Finally, in some complex risks such as cyberbullying, where, besides limiting behaviors, communication and awareness between parents and children are essential, the most effective strategy is to combine both RM and EM.

## Supporting information

**S1 Checklist. PRISMA_2020_checklist_PM_PONE__05_07.**
(DOCX)

**S2 Checklist. JBI critical appraisal checklist.**
(XLSX)

**S1 File. Application_of_eligibility_criteria_with_duplicates.**
(XLSX)

**S2 File. Application_of_eligibility_criteria_without_duplicates.**
(XLSX)

## Author Contributions

**Conceptualization:** Juan Manuel Machimbarrena, Joaquín González-Cabrera.

**Data curation:** David Sevilla-Fernández, Adoración Díaz-López, Joaquín González-Cabrera.

**Formal analysis:** David Sevilla-Fernández, Adoración Díaz-López.

**Funding acquisition:** Joaquín González-Cabrera.

**Investigation:** David Sevilla-Fernández, Adoración Díaz-López, Jessica Ortega-Barón, Joaquín González-Cabrera.

**Methodology:** David Sevilla-Fernández, Adoración Díaz-López, Vanessa Caba-Machado.

**Project administration:** Joaquín González-Cabrera.

**Resources:** David Sevilla-Fernández, Joaquín González-Cabrera.

**Supervision:** Juan Manuel Machimbarrena, Joaquín González-Cabrera.

**Validation:** Vanessa Caba-Machado, Jessica Ortega-Barón, Joaquín González-Cabrera.

**Writing – original draft:** David Sevilla-Fernández, Adoración Díaz-López.

**Writing – review & editing:** David Sevilla-Fernández, Adoración Díaz-López, Vanessa Caba-Machado, Juan Manuel Machimbarrena, Jessica Ortega-Barón, Joaquín González-Cabrera.

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
