## [Decision Letter · Decision Letter 0]

28 Aug 2024

PONE-D-24-27640Parental mediation and the use of social networks: A systematic reviewPLOS ONE

Dear Dr. González-Cabrera,

Thank you for submitting your manuscript to PLOS ONE. After careful consideration, we feel that it has merit but does not fully meet PLOS ONE’s publication criteria as it currently stands. Therefore, we invite you to submit a revised version of the manuscript that addresses the points raised during the review process.

We look forward to receiving your revised manuscript.

Kind regards,

Mohamed Amine Baba

Academic Editor

PLOS ONE

**Journal Requirements:**

This study was funded by Universidad Internacional de La Rioja

[(UNIR Research Plan (2022-2024)] & ITEI BC 170. Ministerio de Ciencia, Innovación y

Universidades-Spain (Pending).

Reviewers' comments:

Reviewer's Responses to Questions

**Comments to the Author**

1. Is the manuscript technically sound, and do the data support the conclusions?

Reviewer #1: Yes

Reviewer #2: Yes

2. Has the statistical analysis been performed appropriately and rigorously? 

Reviewer #1: N/A

Reviewer #2: No

3. Have the authors made all data underlying the findings in their manuscript fully available?

Reviewer #1: Yes

Reviewer #2: Yes

4. Is the manuscript presented in an intelligible fashion and written in standard English?

Reviewer #1: Yes

Reviewer #2: Yes

5. Review Comments to the Author

**Reviewer #1:** First of all, I would like to congratulate the authors for their choice to address this very relevant topic, which concerns the use of social media and its impact on children and adolescents.

The remarks I make aim to improve the work and do not affect the quality of the efforts made.

1- For the abstract:

-You should mention the sources used to identify the included studies.

-You should specify the methods used to assess the risk of bias in the included studies.

-Please specify the main source of funding for the review.

-please indicate the name and registration number of your protocol.

3- In the section "Methodology":

- Have you limited your search to just two languages? This could indeed have an impact on your results if there are other studies published in different languages?

- In your PRISMA flow diagram, please add the criteria used by the research group to exclude articles (phase: eligibility).

-check the total number of items excluded from the “screening” section in your PRISMA flow diagram (163+6+46+12+33= 260).

4- in the "Results" section:

- Please provide the results of your qualitative assessment for each study.

5- In the discussion section:

-It is requested that a paragraph explaining the purpose of this systematic review be included at the outset of the discussion section.

**Reviewer #2: **

Dear authors,

I would like to commend you on the significant work you've done in conducting this systematic review. The topic of parental mediation in adolescents' and children's use of social networks is both timely and highly relevant, given the pervasive role that social media plays in the lives of young people today. Your adherence to the PRISMA protocol and the thorough analysis of the literature from the past decade provide valuable insights into this critical area. The findings you have presented offer a strong foundation for future research and practical interventions.

Now, I would like to offer a few suggestions to further strengthen your manuscript, particularly in the methodology section.

**Eligibility criteria:**

Please add a section detailing the "Eligibility criteria" in your methodology. It would be beneficial to present this information in a table format under the heading "Eligibility criteria according to the PICOS framework."

**Justification for the time period (2012-2023):**

It is important to justify why you selected the period between 2012 and 2023 for your review. Please include a rationale for this choice in the methodology section.

**Flow diagram discrepancy:**

Please review the number of articles excluded in your study, as there seems to be an issue with the totals in your flow diagram. Ensure to explain any discrepancies.

**Risk of bias:**

For assessing the risk of bias, I recommend calculating the Kappa statistic to evaluate the level of agreement between reviewers.

**Type of study:**

The type of study was not mentioned in the characteristics table of the included studies. Please add this information to the table.

**Sex ratio:**

Instead of providing the percentage of females in the characteristics table, I suggest using the sex ratio to give a clearer representation.

**Table 3 (Results section):**

To make your results more scientifically robust, Table 3 should include percentages, statistical test results, and other relevant metrics that highlight which type of mediation strategy is more effective.

6. PLOS authors have the option to publish the peer review history of their article (what does this mean?). If published, this will include your full peer review and any attached files.

Reviewer #1: **Yes: **Dr. ARECHKIK ABDERRAHMAN

Reviewer #2: **Yes: **Mohamed Amine BABA

---

## [Author Response · Author response to Decision Letter 0]

11 Sep 2024

Reviewer #1: First of all, I would like to congratulate the authors for their choice to address this very relevant topic, which concerns the use of social media and its impact on children and adolescents. The remarks I make aim to improve the work and do not affect the quality of the efforts made.

Authors  Dear Prof. Dr. Arechkik Abderrahman, in the first place we would like to truly thank you for the time and the effort you have dedicated to review our manuscript. Secondly, we really appreciate the positive words you dedicate to our work.

1- For the abstract:

-You should mention the sources used to identify the included studies.

-You should specify the methods used to assess the risk of bias in the included studies. 

-Please specify the main source of funding for the review.

-please indicate the name and registration number of your protocol.

Authors  Thank you for your comments. Following your suggestion, we have added to the abstract the information related to the sources of the studies, the risk of bias method and the number of our Prospero’s protocol. However, even if we appreciate your suggestion about including the source of funding, we do not consider entirely appropriate or necessary to do so. In this sense, according to the journal's rules for authors the abstract should focus solely on describing the main objective(s) of the study, explaining how the study was conducted, without methodological detail and summarizing the most important results and their significance. The funding information has already been added on the PONE web platform so that it can be known in the final version of the manuscript. Below we present the abstract with the requested modifications: 

Social networks are used daily by almost all adolescents and young people. They are used extensively, and their positive aspects are well-known, but they present multiple risks. Parents' mediation (PM) in their children's use of social networks (SNs) to prevent numerous problems has been recently researched. This systematic review analyzed works published since 2012 about online PM strategies concerning the use of SNs by children/adolescents aged between 9 and 18 years old. Following the PRISMA protocol, this review was submitted to Prospero (ID: CRD42022345033), five electronic databases were examined (WOS, SCOPUS, ERIC, ProQuest Psychology, and PubMed), and to assess the risk of bias the ROBIS tool was used. 32 papers were selected. The results indicate that PM strategies (especially the restrictive strategies over the enabling ones) effectively reduce the time spent on SNs and their associated risks. However, it is unclear whether one type of strategy is more effective or whether a combination of the different strategies, depending on the problem to be addressed, is most effective. Finally, no studies have been found that have analyzed the relationship between PM strategies and the benefits associated with the use of SNs. Possible lines of action for future programs and research are proposed.

3- In the section "Methodology":

- Have you limited your search to just two languages? This could indeed have an impact on your results if there are other studies published in different languages?

Authors  Yes, you are right in your statement, and we fully agree with you. So, this aspect was already justified in the list of limitations of our manuscript:” Finally, only studies in English and Spanish were included in this review, so papers of interest in other languages may have been excluded”. However, only 6 papers have been eliminated by language (see Figure 1). Language limitations are common in these studies, since they are limited to the linguistic competence of the researchers themselves.

- In your PRISMA flow diagram, please add the criteria used by the research group to exclude articles (phase: eligibility).

Authors  Thank you for your suggestion. We have included it. 

-Check the total number of items excluded from the “screening” section in your PRISMA flow diagram (163+6+46+12+33= 260).

Authors  Thank you very much for your appreciation. Thanks to it we realized that it was a mistake, and that the sum was incorrect. We have modified it. 

4- in the "Results" section:

- Please provide the results of your qualitative assessment for each study.

Authors  Following your suggestion, we have added the following information to the result section: 

.. As for the quality assessment, 90,6% of the studies presented high quality, and 9,4% medium quality, no studies were of low quality (see Table 3).

5- In the discussion section:

-It is requested that a paragraph explaining the purpose of this systematic review be included at the outset of the discussion section.

Authors  We agree with you, so we have added the following information to the discussion section:

In this systematic review, we intended to go a step further, providing systematized knowledge about the possible evidence of the use of SNs and its relationship with the different PM strategies. Specifically, we focused on analyzing the relationship between families' PM and their children's usage time of SNs, the effects that different PM strategies have on reducing the risks of SNs, and how different PM strategies are relate to the benefits associated with SNs.

 ------------------------------------------------------------------------------------------------------------------------------------------------------------------------------

Reviewer #2: 

Dear authors,

I would like to commend you on the significant work you've done in conducting this systematic review. The topic of parental mediation in adolescents' and children's use of social networks is both timely and highly relevant, given the pervasive role that social media plays in the lives of young people today. Your adherence to the PRISMA protocol and the thorough analysis of the literature from the past decade provide valuable insights into this critical area. The findings you have presented offer a strong foundation for future research and practical interventions.

Now, I would like to offer a few suggestions to further strengthen your manuscript, particularly in the methodology section.

Authors  Dear Prof. Mohamed Amine, in the first place we would like to truly thank you for the time and the effort you have dedicate to review our manuscript. Secondly, we really appreciate the positive words you dedicate to our work.

Eligibility criteria:

Please add a section detailing the "Eligibility criteria" in your methodology. It would be beneficial to present this information in a table format under the heading "Eligibility criteria according to the PICOS framework."

Authors  Following your suggestion, we have created a table (Table 1) in our methodology section including the eligibility criteria of our study according to PICOS Framework. Please refer to the attached document for the table (or manuscript). The PONE application only allows us to include plain text.

Justification for the time period (2012-2023):

It is important to justify why you selected the period between 2012 and 2023 for your review. Please include a rationale for this choice in the methodology section.

Authors  Determining the exact year in which social networks began to be used on a massive scale and in which the first research on online parental mediation began to be developed at the same time is not an easy task. We therefore decided to set this time period after the following reflection: In the first place, the use of social networks on smartphones began around 2010, this period coincides with the increasing adoption of smartphones by all the population, and the availability of the first mobile applications for social networks such as Facebook and Twitter. However, the widespread and massive use of social networks started in 2012 with the creation of Instagram and later, in 2016, Tik Tok (Feldkamp, 2021). While there were previous social networks (such as Facebook), the main users of it are not the teenagers from the post-2020 studies, although they are from the studies around 2012. However, the use was not as massive (since this occurred with the massive use of smartphones and the creation of high-speed mobile networks).

On the other hand, it was not until beginning of 2012, when online parental mediation become a line of interest for scientist and started to be investigated in relation with other variables, apart from some previous approximation (mainly theorical) made by Livingstone & Helsper (2008) and the first Global Kids and EU Kids studies which were conducted in the early 2010s.

Selecting the lower limit of a time range is complicated and arbitrary (we are aware of this). Other decisions could have been made, but we believe that this decision is rational and in line with the development of the study constructs.

In any case, these are more than 10 years (and the last ones), so we believe that the review is able to gather the most relevant milestones of the scientific literature.

We have added to the methodology section the following information:

[…] Determining the exact lower limit of a time range is complicated and arbitrary. We therefore decided to set this period because the widespread and massive use of social networks between teenagers and young adults started in 2012 with the creation of Instagram and later, in 2016, Tik Tok (Feldkamp, 2021). Furthermore, it was not until beginning of 2012, when online parental mediation become a line of interest for scientist and started to be investigated in relation with other variables (Livingstone & Helsper, 2008). 

And to the references list, the following references: 

Feldkamp, J. (2021). The rise of TikTok: The evolution of a social media platform during COVID-19. Digital responses to Covid-19: Digital innovation, transformation, and entrepreneurship during pandemic outbreaks, 73-85.

Livingstone, S., & Helsper, E. J. (2008). Parental mediation of children's internet use. Journal of broadcasting & electronic media, 52(4), 581-599.

Flow diagram discrepancy:

Please review the number of articles excluded in your study, as there seems to be an issue with the totals in your flow diagram. Ensure to explain any discrepancies.

Authors  Thank you very much for your appreciation. Thanks to it we realized that it was a mistake, and that the sum was incorrect. The sum was 268, but the correct result is 260. We have modified it in the flow diagram. We have taken advantage of this error to review the rest of the procedure, and we have identified another mistake in the document uploaded to the repository called: "Aplicación_criterios_elegibilidad_sin_duplicados.xls" regarding an article that was listed as included when it was not. We have solved it. 

We are grateful that thanks to the revision these details can be corrected.

Risk of bias:

For assessing the risk of bias, I recommend calculating the Kappa statistic to evaluate the level of agreement between reviewers.

Authors  According to your suggestion we have calculated the Kappa statistic, and we have added the following information to the Risk of bias section: 

[…]and the Cohen´s Kappa Index was 0,963

Type of study:

The type of study was not mentioned in the characteristics table of the included studies. Please add this information to the table.

Authors  All the studies were observational, so we have preferred to comment on this in section 3.1. Other information of interest is included in Table 3 and in the text.

 […] all of them was observational studies

Sex ratio:

Instead of providing the percentage of females in the characteristics table, I suggest using the sex ratio to give a clearer representation.

Authors  We have added the sex ratio to table 3. We consider that including this aspect has improved the quality of our work, thank you very much for your suggestion. 

Table 3 (Results section):

To make your results more scientifically robust, Table 3 should include percentages, statistical test results, and other relevant metrics that highlight which type of mediation strategy is more effective.

Authors  We agree with you in the fact that including in our results percentages and statistical test results would make our study more robust. However, we believe that due to the plural statistical analysis that have been use in the different studies, offering this information in a summary way in the table could be problematic and led the reader to a mistake or misunderstanding. Furthermore, this study is not a meta-analysis, so despite we have found your suggestion very interesting, the authors goal was just to know and show the association between the variables addressed. 

We believe that Table 3 is currently clear and presents simple and easily interpretable information for researchers. We are confident that those who need more information (or require meta-analysis) will go to the original source. In this approach, the authors prioritize parsimony and ease of interpretation.

Nevertheless, we highly appreciate your comment, and we will have this point into account for future studies.

---

## [Editor Report · Decision Letter 1]

30 Sep 2024

**Parental mediation and the use of social networks: A systematic review**

**PONE-D-24-27640R1**

Dear Dr. Joaquín González-Cabrera

We’re pleased to inform you that your manuscript has been judged scientifically suitable for publication and will be formally accepted for publication once it meets all outstanding technical requirements.

Within one week, you’ll receive an e-mail detailing the required amendments. When these have been addressed, **you’ll receive a formal acceptance letter and your manuscript will be scheduled for publication*****.***

Kind regards,

Mohamed Amine Baba

Academic Editor

PLOS ONE
---

## [Editor Report · Acceptance letter]

24 Oct 2024

PONE-D-24-27640R1 

PLOS ONE

Dear Dr. González-Cabrera, 

I'm pleased to inform you that your manuscript has been deemed suitable for publication in PLOS ONE. Congratulations! Your manuscript is now being handed over to our production team.

Kind regards, 

on behalf of

Pr Mohamed Amine Baba 

Academic Editor

PLOS ONE